# ERK1/2: An Integrator of Signals That Alters Cardiac Homeostasis and Growth

**DOI:** 10.3390/biology10040346

**Published:** 2021-04-20

**Authors:** Christopher J. Gilbert, Jacob Z. Longenecker, Federica Accornero

**Affiliations:** Department of Physiology and Cell Biology, Dorothy M. Davis Heart and Lung Research Institute, The Ohio State University, Columbus, OH 43210, USA; Christopher.Gilbert@osumc.edu (C.J.G.); Jacob.Longenecker@osumc.edu (J.Z.L.)

**Keywords:** ERK, extracellular matrix, heart, cardiac hypertrophy

## Abstract

**Simple Summary:**

Understanding how cardiac cells respond to external stimuli is essential for developing interventions that mitigate pathologies of the heart. Therefore, in this review, we summarize critical knowledge related to a key molecular pathway that mediates cellular responses.

**Abstract:**

Integration of cellular responses to extracellular cues is essential for cell survival and adaptation to stress. Extracellular signal-regulated kinase (ERK) 1 and 2 serve an evolutionarily conserved role for intracellular signal transduction that proved critical for cardiomyocyte homeostasis and cardiac stress responses. Considering the importance of ERK1/2 in the heart, understanding how these kinases operate in both normal and disease states is critical. Here, we review the complexity of upstream and downstream signals that govern ERK1/2-dependent regulation of cardiac structure and function. Particular emphasis is given to cardiomyocyte hypertrophy as an outcome of ERK1/2 activation regulation in the heart.

## 1. Introduction

Within the heart, cardiomyocytes represent the primary working cell responsible for contraction. However, how cardiomyocytes function is highly dependent upon the extracellular environment. In the extracellular matrix, a plethora of signals converge to communicate cellular needs and allow for stress responses [1]. In addition to serving as a signaling hub, the mechanical properties of the extracellular matrix itself can dictate cardiomyocyte behaviors [2]. Signal integration is necessary for survival, and as such, cardiomyocytes express a variety of cell membrane receptors that allow for signal transduction. Within these, integrins can directly sense mechanical deformations driven by changes in the extracellular matrix composition and stiffness [3]. On the other hand, other important classes of receptors such as the G-protein-coupled and the tyrosine kinase ones (GPCRs and RTKs respectively) are modulated by select extracellular factors that can be secreted by different cardiac cell types or even reach the heart from distal place while circulating into bloodstream [4].

Regardless of the origin of the extracellular signal, cardiomyocyte responses are dependent upon intracellular effectors. Extracellular signal-regulated kinases 1 and 2 (ERK1/2 or ERK) are effectors of primary importance for their role as integrators of signals derived from various extracellular events. Indeed, ERK can be activated by integrins, GPCRs and RTKs, and even more importantly ERK is critical for signaling crosstalks between all these receptors [5]. Therefore, it is not surprising that ERK plays a pivotal role in regulation of cardiomyocyte fate. While ERK acts throughout development, downstream of both acute and chronic stress, and in response to multiple types of stressors, the regulation of cardiomyocyte size and shape through hypertrophic remodeling has been highlighted as an indisputable role for ERK in the heart [6]. Therefore, this review will particularly focus on cardiac hypertrophy by describing the mouse models and studies that revealed the importance of the ERK pathway in cardiomyocytes. Prior to describing the consequences of ERK activation in the heart, we will describe the general mode of ERK signaling with a particular emphasis on the known downstream targets that ultimately mediate ERK actions.

## 2. The ERK1/2 Signaling Module

Mitogen-activated protein kinase (MAPK) cascades constitute serial phosphorylation networks by which external stimuli beget cellular consequences. To date, three such pathways have been established: extracellular signal-regulated protein kinase (ERK), c-Jun N-terminal kinase/stress-activated protein kinase (JNK/SAPK), and p38 kinase. Each of these comprises a tiered module whereby an MAPK kinase kinase (MAP3K) activates an MAPK kinase (MAP2K), which, in turn, activates an MAPK [7]. ERK signaling is initiated at the membrane through a milieu of receptor proteins, such as RTKs, GPCRs, and integrins. Classically, ligand binding elicits the recruitment of adaptor proteins and exchange factors to yield activated GTP-Ras, to which cytosolic kinases Raf-1, B-Raf, and A-Raf (MAP3Ks) are recruited [5]. Activated Raf functions to propagate the signal through the activation of dual-specificity kinases MEK1 and MEK2 (MAP2Ks), which function to phosphorylate ERK1 and ERK2 (MAPKs) [5]. Beyond these canonical factors, other MAP3K signaling components have also been observed to activate ERK1/2 under specific circumstances, such as MAP3K8 (also known as Tumor progression locus 2 or TPL2) [8] and MAP3K3 (also known as MEK kinase 3 or MEKK3) [9].

MAPKs are phosphorylated at Threonine (Thr) or Tyrosine (Tyr) residues within a Thr-X-Tyr motif in the kinase domain [10]. After their phosphorylation, ERK1 and ERK2 (henceforth referred to as singular ERK) function as kinases with redundant functions and regulation under most conditions. ERK functions as a proline-directed protein kinase (PDPK), whereby phosphorylation occurs to Ser/Thr residues adjacent to a Pro, to activate targets tied to cellular growth and proliferation [5]. ERK is ubiquitously expressed in mammals, highlighting its importance in regulating cell signaling in response to stimuli. To date, hundreds of proteins have been identified to interact with ERK. The fidelity of these interactions is arbitrated by targeting the consensus sequence PXS/TP and via specialized binding motifs [11]. ERK activity is also controlled via several phosphatases to regulate the duration of the transduced signal. While the dephosphorylation of ERK may be achieved by Ser/Thr [12] or Tyr phosphatases [13], there also exist specialized dual specificity phosphatases (DUSPs) that function to dephosphorylate both residues concurrently. These DUSPs comprise a phosphatase domain with a conserved catalytic site with Asp, Cys, and Arg residues within a flexible enzymatic pocket to accommodate phosphorylated targets [14]. Cytoplasmic DUSP6 and DUSP7 have been identified as ERK-specific, as has inducible nuclear DUSP5 [15,16]. Nuclear DUSP1, DUSP2, DUSP4, and DUSP9 also regulate ERK signaling, among other MAPK modules [16,17,18]. ERK is further regulated through the feedback of other MAPK components, within the ERK compartment and other modules, and several studies are aimed at better understanding the intersection of these conserved cascades [19]. Crosstalk with other cellular pathways such as PI3K–AKT–mTOR [20] and NF-κB [21] has also been identified.

Among the identified cytoplasmic targets of ERK are cytoskeletal elements, membrane receptors, and adherens junction proteins (Table 1). Several nuclear targets of ERK have also been identified (Table 1). Under resting conditions, ERK is retained in the cytoplasm by MEK1/2 or the microtubule network [22]. Upon its phosphorylation, ERK dissociates from MEK1/2, and phospho-ERKs have been observed to dimerize prior to interaction with cytoplasmic substrates. The nuclear translocation of ERK is mediated by passive diffusion of the monomer or active transport of the dimer as well as interaction with the nuclear pore complex [22]. Upon translocation, ERK directly phosphorylates transcription factors, such as Elk-1, to modulate cell growth [5]. Interestingly, such nuclear modifications solely occur by way of the monomer [23]. The spatial fidelity of interaction is further enforced by scaffold proteins to which ERK dimers may remain bound to prevent ERK nuclear translocation and favor its targeting of cytosolic substrates [23]. ERK also functions to phosphorylate mitogen-activated protein kinase-activated protein kinases (MAPKAPs). Activation of MAPKAPs RSK1 (ribosomal s6 family kinase 1), RSK2, and RSK3 elicits their nuclear translocation and downstream phosphorylation of transcription factor cAMP response element-binding protein (CREB) to promote cell survival [24]. ERK also phosphorylates other MAPKAPs as MSK1/2 (mitogen- and stress-activated protein kinase), which are mediators of gene activation via histone H3 phosphorylation, and MNKs (MAP kinase-interacting serine/threonine-protein kinase), which play a role in phosphorylating translational machinery and splicing factors [25]. ERK has also been shown to phosphorylate and inactivate proapoptotic proteins, while activating proteins tied to transcription and translation [5]. Generally, the ability of ERK to control gene expression as well as its direct cytoplasmic role in the regulation of cell architecture defines this kinase as a central mediator of cell size and shape with clear ramifications for cardiomyocyte hypertrophy and remodeling. To understand how this is ultimately achieved, we will first describe the upstream extracellular signals that lead to ERK activation in the heart.

## 3. Sensing of Extracellular Stimuli That Modulate ERK1/2 Activity in Cardiomyocytes

As the name “extracellular signal-regulated kinase” suggests, ERK is a key intracellular transducer of extracellular signals. Therefore, it is essential to discuss how the extracellular signals are parlayed to elicit cellular changes. At the interface between the extracellular milieu and ERK, there are three major classes of membrane receptors: integrins, G-protein-coupled receptors, and tyrosine kinase receptors. Ultimately, ERK is the coordinator of multiple signaling events that are initiated by the receptors mentioned above, and we will therefore describe the main characteristics of these cellular pathways.

### 3.1. Integrins

The first of the three types of membrane-bound receptors that modulate ERK signaling in cardiomyocytes to be discussed are the integrins, which are a large class of structurally related proteins that have been shown to be integral to many cellular processes. A key characteristic of these receptors is their involvement in mechanotransduction, which consists of sensing the mechanical deformation of structural elements, including the extracellular matrix (ECM) and intracellular cytoskeletal components, and converting these changes into a signal that can alter cell morphology and behavior. The cellular response to stretch is highly linked to cardiac hypertrophy, highlighting the importance of integrins in the heart. As pressure increases in the heart, cardiomyocytes become elongated, and components of the ECM and cytoskeleton are mechanically altered. This change in structure is partially relayed to the nucleus via integrins.

While integrins have been extensively studied in many different cellular contexts, the majority of studies in cardiomyocytes that focus on integrin-dependent regulation of ERK signaling has described hypertrophy and cell survival as the main downstream cellular responses. In cultured adult feline cardiomyocytes, the activation of beta 3 integrins using RGD (Arg-Gly-Asp)-containing peptides led to hypertrophy through the regulation of p70S6 kinase (S6K1) and activation of ERK [69]. Later, in 2005, an Italian group linked integrin signaling to ERK-mediated cardiac hypertrophy, where they showed that the overexpression of an integrin binding protein, melusin, was protective by inducing adaptive hypertrophic remodeling following pressure overload-induced heart failure in transgenic mice. The authors noted that multiple signaling pathways were activated, including ERK, but the specific contribution of ERK to this response was not tested, for instance, through the use of an ERK inhibitor [70]. Another direct link between integrin signaling and hypertrophy is a study that induced hypertrophy in rat cardiomyocytes using endothelin-1 (ET-1) and showed that under these conditions, beta 1 integrin expression and reactive oxygen species (ROS) production are increased. Upon blockade of integrin signaling, the hypertrophic response to ET-1 was blunted, indicating a critical role for integrins in this response. Interestingly, it was shown that ROS-induced ERK activation is responsible for the upregulation of beta 1 integrin, implying that in this particular case, ERK activation was not a consequence of beta 1 integrin upregulation, but rather, the cause [71]. More recently, Nakamura et al. highlighted a connection between integrin signaling and ERK activation in cardiomyocytes using a transgenic mouse with knock out of a disintegrin and metalloprotease 12 (ADAM12). Here, they showed that loss of ADAM12 increased beta 1 integrin abundance with concomitant increase in ERK activation, although the authors did not assess if the activation of ERK was directly related to the upregulation of beta 1 integrin [72]. Similarly, the expression of microfibrillar associated protein 4 (MFAP4), another ECM protein, has been shown to directly activate RGD-binding integrin signaling in cardiomyocytes and favor adaptation to stress [73].

There are also several papers that focused specifically on the role of integrins in ERK activation during stretch of cardiomyocytes. The earliest of these papers explored the connection between brain natriuretic peptide (BNP) and cardiomyocyte stretch in vitro, which is a situation in which BNP and ERK are activated. These authors utilized two integrin binding proteins, RGD peptides and fibronectin, to explore how integrins are involved in the upregulation of BNP in response to stretch. After showing that treatment with these proteins induces BNP similar to stretch, they blocked several integrin isoforms (beta1, beta3, and alpha5beta5) and found that stretch-induced BNP expression in cardiomyocytes was indeed integrin-dependent [74]. This critical role for integrins in detecting mechanical changes to the cell is further underscored by the finding that stretch of neonatal rat ventricular cardiomyocytes activates ERK via beta 1 integrin [75]. Together, these findings highlight a critical role for integrin signaling in detecting changes to the mechanical structure of cardiomyocytes during hypertrophy and stretching.

Overall, the above findings provide strong evidence to support the idea that integrin-induced ERK activation is critical to the maintenance of cardiomyocyte homeostasis. Although in many of these cases, different compounds were used in the presence of different stressors, one central theme emerged: without integrin signaling, the cardiomyocyte would be unable to respond to cardiac stress.

### 3.2. G-Protein-Coupled Receptors

The second of the three receptor types to be discussed are the GPCRs, which are a family of membrane bound receptors, each of which contain an N-terminal extracellular domain, seven transmembrane a-helices, and a C-terminal cytoplasmic domain. Critical regulators of cardiomyocyte hypertrophy and function, such as endothelin I, angiotensin II, isoproterenol, lysophosphatidic acid, and phenylephrine, act through GPCRs, greatly underscoring the importance of these ERK-activating receptors in the heart [76,77,78,79,80,81,82,83]. A wide range of mechanisms for how GPCR activation can modulate ERK activation have been highlighted, such as a requirement for ROS [76,84], protein kinase C [80], cAMP and PKA [81], and even RTK transactivation [80]. The sheer breadth of potential GPCR activators and their many downstream effectors described here is just barely indicative of the complexity of these pathways and their regulation. Adding to this complexity is that multiple G protein isoforms can be expressed in different conditions, and any one of those individual g protein subunits could be differentially regulated, greatly influencing what effectors are responsible to carry the signal forward. There are many studies that have focused on studying the differential contributions by individual G protein subunits and their isoforms, frequently through the use of genetically engineered cells or animals. For instance, several groups have examined hypertrophic effects attributed specifically to the Gaq subunit via its specific activation or overexpression, showing that its activation alone is sufficient to produce a robust hypertrophic phenotype with concomitant ERK activation that leads to heart failure [85,86,87,88]. There are three studies of note that have pointed out that the active GTP-bound alpha subunit is not the only potential effector of GPCR activation. Although one study was focused on the beta/gamma subunit released by the Gs alpha subunit [89] and the other two focused on that which was released by the Gq alpha subunit [87,90], all three concluded that the released beta/gamma subunit is capable of binding to ERK, modulating its activity. It is lastly worth mentioning proteins that contain regulator of G protein signaling (RGS) domains, which are involved in reducing the effect of GPCR activation. Upon the introduction of the RGS domain containing proteins to cardiomyocytes, there is a block in ERK activation, and the response to hypertrophic stimuli is reduced [91,92]. The opposite outcome occurs when RGS signaling is reduced and Gq-mediated ERK activation is increased [93]. Other groups have assessed the role of specific g protein alpha subunit isoforms in RGS mediated ERK activation [94] as well as the role of different RGS domains in activating ERK [95]. These results indicate that RGS protein domains may be viable targets in the treatment of diseases that involve dysregulated hypertrophic responses, such as heart failure. Regardless of the method used to induce hypertrophy and the specific mechanism that ultimately leads to its development, there is a large body of evidence supporting the notion that GPCR-induced activation of ERK is absolutely essential to maintaining the proper hypertrophic response within cardiomyocytes.

Finally, because adrenergic receptors (AR) are GPCRs, several studies have specifically focused on adrenergic signaling and its effect on ERK activation, many of which have been described above. Two more worth noting have focused on the subcellular localization of either ERK or the receptors, finding that alpha1AR activation alters the localization of active ERK to caveolae in a fashion not dissimilar from that described for the S1P receptor above [96], and that alpha1AR compartmentalization itself can regulate ERK activation [97]. These two studies highlight yet another intricate form of regulation for the activation of ERK by GPCRs in cardiomyocytes and further underscore the sheer complexity of signal transduction events that occur between GPCR stimulation and eventual activation of ERK, with more undoubtedly yet to be uncovered.

### 3.3. Receptor Tyrosine Kinases

In addition to integrins and G-protein coupled receptors, receptor tyrosine kinases (RTK) are critical mediators in the process by which cells detect extracellular signals, convert the signals into a change in ERK signaling, and modulate cellular processes in response. Classically, RTKs were discovered as being receptors for growth factors that play a role in cellular proliferation, with the initial connection between tyrosine phosphorylation and this outcome found during addition of epidermal growth factor (EGF) to epidermal cells [98], and, later, insulin to lymphocytes [99] and platelet-derived growth factor (PDGF) to glial cells and fibroblasts [100]. Since then, many more RTKs have been identified and studied, many of which are involved in processes beyond just proliferation. Despite several signaling cascades that can originate upon RTKs activation, the MAPK cascade is certainly recognized as an important downstream mediator of these receptors. We will particularly focus our discussion on key studies reporting RTK-dependent ERK activation in cardiomyocytes.

Of the studies that focused on the role of RTKs in the activation of ERK during cardiac hypertrophy, the majority utilized primary cultured rat cardiomyocytes to test crosstalk between GPCRs and RTKs. Yamazaki and colleagues explored the hypertrophic effect of endothelin-1 and noted an RTK’s contribution to endothelin-driven and ERK-dependent cardiomyocyte hypertrophy [79]. Another study assessed the contribution of PDGF RTK as a mediator of cardiomyocyte hypertrophy downstream of another GPCR activator, angiotensin II, where ERK activation led to phosphorylation of the PDGF receptor, showing another example of RTK-GPCR crosstalk for coordination of cardiomyocyte growth [77]. Further building upon the concept of crosstalk between GPCRs and RTKs is the study by Carrasco et al., which utilized cultured rat cardiomyocytes to assess how G proteins are critical to the induction of hypertrophy by insulin-like growth factor 1 (IGF-1) [101]. IGF-1 had previously been shown to activate ERK through the IGF-1R, which is an RTK. Upon the blockade of G-protein signaling using pertussis toxin or BARKct, ERK activation by IGF-1 treatment was abolished, indicating an essential role for the crosstalk between RTKs and G-proteins in the activation of ERK in cardiomyocytes by IGF-1 [101]. A testament to the critical role for RTK signaling in cardiac hypertrophy is the finding by Yu et al. that the activation of ERK by RTKs is conserved, even in the drosophila heart [102]. Using transgenic flies, the authors showed how overexpression of the RTK EGFR led to cardiac hypertrophy and how this effect was prevented by blocking ERK activation [102]. Together, these results highlight a role for EGFR activation and subsequent signaling through ERK in the development of cardiac hypertrophy.

RTKs have been greatly studied since their initial discovery, with papers on the topic reaching into the tens of thousands. Despite this, knowledge of the role that they play in transmitting signals to ERK within the cardiomyocyte is severely lacking. The studies that have focused on this have shown that RTK signaling in cardiomyocytes is clearly critical to the development of hypertrophy as well as in the defense against stress, as evidenced by the requirement for RTK and ERK activation in each context examined thus far. Although RTK signaling is not usually the sole contributor to the regulation of a specific cellular process, be it hypertrophy, apoptosis, or proliferation, its contribution to the maintenance of cardiomyocyte homeostasis cannot be understated.

## 4. Integration of ERK1/2 Signaling for Cardiomyocyte Homeostasis and Stress Responses

As discussed above, receptor signals are transduced to beget cardiac ramifications. Indeed, pressure overload stimulates ERK activity by way of integrins and GPCRs [69,70,74,75,92]. RTKs are also essential to cardiac homeostasis and stress response [77,79,101,102,103,104,105,106]. Numerous studies have explored the roles of canonical ERK module components in both physiological and pathological hypertrophy with clear evidence of the impact of ERK activation for regulation of gene expression. Indeed, several transcription factors, such as archetypical effectors Elk-1 [53] and Ets1 [55], have been identified downstream of ERK in the cardiomyocyte. The direct phosphorylation of transcription factor GATA4 at serine 105 has been identified as necessary to hypertrophic response to pressure overload or adrenergic stimulation by phenylephrine [107,108], where phosphorylation may be facilitated by either ERK or p38 [108]. Genetic ablation of GATA4 precludes hypertrophy, and this phenotype was not found to be rescued in activated MEK1 transgenic mice, suggesting GATA4 as a critical mediator of the MEK1–ERK pathway in the heart [108]. Recently, ERK effector RSK3 was identified to phosphorylate another transcription regulator, serum-responsive factor (SRF), where downstream signaling modulates cardiomyocyte growth in width, which is essential to concentric hypertrophy [12]. Recent reports also indicate crosstalk between ERK and transcription factor STAT3 signaling to mediate cardioprotection in myocytes treated with doxorubicin, again highlighting the importance of the ERK pathway for the integration of stress responses in the heart [109]. Below, we will describe the key studies that definitively linked ERK and its regulation to the control of cardiomyocyte size, shape, and function.

### 4.1. Modulation of ERK Kinase Cascade in Cardiomyocyte

Upstream of ERK, RAF is essential to hypertrophic responses, and dominant negative RAF mice subjected to pressure overload via transverse aortic constriction fail to develop adaptive hypertrophy and show increased cardiomyocyte death even without stress [110]. RAF’s target MEK has also extensively been studied in the heart. Mice overexpressing MEK1 showed an increase in cardiac wall thickness and improved cardiac function, which correlated to an increase in ERK phosphorylation and total ERK levels [111]. These mice were also less prone to apoptosis induced by ischemia/reperfusion [112]. Importantly, transgenic mice with constitutively active ERK also show increased ventricular contractility under pressure overload [6]. In loss-of-function studies, ERK2 ablation was observed to be lethal and heterozygosity significantly reduced fractional shortening following transverse aortic constriction-induced pressure overload compared to WT mice. ERK1-null mice showed no significant change in hypertrophy but trended toward reduced cardiac function [113]. These data suggest a role for ERK in concentric hypertrophy. To better understand the role of ERK signaling in balancing hypertrophy, a study by Kehat and colleagues found that while single knockout showed no phenotype at baseline, double knockout elicited cardiomyocyte elongation to yield eccentric hypertrophy, suggesting a role for ERK1/2 signaling in maintaining a physiological cardiomyocyte geometry [114]. Cardiomyocytes from these mice were able to undergo hypertrophic growth in width upon angiotensin II or phenylephrine administration, suggesting that neurohormonal stimulation can at least in part bypass ERK signaling despite the requirement for ERK to prevent eccentric cardiomyocyte remodeling [114]. These data have been corroborated by pressure overload studies, where upon transverse aortic constriction, mice show an initial increase in ERK phosphorylation, which decreases with functional decline of the heart [115]. In sum, these data suggest a role for ERK in hypertrophy, particularly in preventing eccentric hypertrophy in response to pressure overload stress. Keeping in mind the proven importance of ERK for cardiac homeostasis, we will next describe how scaffold proteins can regulate its activation.

### 4.2. Scaffolds in Regulation of ERK Signaling in the Heart

Specialized scaffold proteins function to spatially and temporally regulate the transduced ERK signal downstream of all main ERK-activating receptors in cardiomyocytes (Figure 1). Downstream of RTK receptors, biologically inactive Ras-GDP requires conversion to Ras-GTP via guanine exchange factors (GEFs) to induce conformational change and enzymatic activity. Indeed, exchange factor son-of-sevenless (SOS) is recruited to RTKs to facilitate this exchange [116]. Scaffold protein Shoc2 has been identified as critical to Ras activation and vital to cardiac development, where ablation yields severe cardiac defects and embryonic lethality [117]. Moreover, a Shoc2 mutation has been identified in Noonan-like diseases, cementing a role for this scaffold in heart development [118]. Overall, both Shoc2 and SOS appeared critical for the RTK-mediated ERK cascade. Additionally, ErbB2-interacting protein (Erbin) has been identified as an antagonist of RTK Erb2 downstream activity, where null mice exhibit pathological hypertrophy and heart failure following pressure overload [119].

Scaffold proteins also play an integral role in integrin signal propagation. Indeed, integrin-binding chaperone protein melusin is essential to adaptive hypertrophy, where expression is upregulated in response to mechanical overload stress [120]. Melusin-null mice show reduced cardiac contractility and dilated cardiomyopathy following pressure overload [121]. Contrarily, the overexpression of melusin has been shown cardioprotective following pressure overload and, following coronary ligation, it reduces the extent of cardiac damage through an enhanced phosphorylation of ERK [70]. Another scaffold protein identified for integrins is motif-containing GTPase-activating protein 1 (IQGAP1), which has roles in cell adhesion and cytoskeletal organization. IQGAP1-null mice more quickly develop eccentric hypertrophy and reduced cardiac contractility due to impaired ERK signaling [122]. Melusin, IQGAP1, and Raf-MEK1/2-ERK1/2 have been observed to coprecipitate in complex with FAK (focal adhesion kinase), suggesting the synergism of these scaffolds to yield integrin-mediated ERK activity. Interestingly, the GTPases attributed to IQGAP1 and canonical ERK signaling were not detected in the complex, suggesting that the interaction does not involve Ras activation [123]. While ERK activity independent of Ras has been previously reported [124], the details of Raf activation through this cascade in the cardiomyocyte still need be elucidated.

For GPCRs, data suggest a role for scaffold β-arrestin in mediating GPCR crosstalk in the ERK module following specific GPCR binding [125]. G-protein-independent activation of beta1AR by β-arrestin has been observed and was found to be cardioprotective. Interestingly, this conferred protection was by way of RTK EGFR [126], suggesting crosstalk between the GPCR and RTK routes. Indeed, RTK agonist Erbin has also been identified as an antagonist to GPCR beta2AR [127].

Further scaffolds have been identified among myofilament-associated proteins four and a half LIM domain protein-1 (FHL1) [128] and ankyrin repeat domain (ANKRD1) [129], which play a role in sarcomeric ERK signaling. These proteins also interact with GATA4, reinforcing the concept of ERK-mediated regulation of gene expression where the balance between cytoplasmic and nuclear ERK can control the nuclear translocation of the transcription factor GATA4 and therefore impact transcription [130,131]. Indeed, loss-of-function studies with ANKRD1 and FHL1 show blunted hypertrophic response. ANKRD1 has been shown to recruit GATA4 and localize to the sarcomere with ERK1/2, after which ANKRD1 nuclear translocation begets the upregulation of hypertrophic genes [129]. As the duration of ERK activation is regulated by phosphatases, we will now describe this important step of signal termination.

### 4.3. Impact of Cardiac ERK Signal Termination by Phosphatases

ERK signaling is further regulated by specialized phosphatases. Indeed, the overexpression of DUSP1 has been observed to suppress MAPK signaling to blunt hypertrophy following pressure overload [132]. Dual specificity kinase 6 (DUSP6) has been shown to specifically dephosphorylate cytoplasmic ERK1/2 in vitro, while DUSP5 has also been identified as a negative regulator of nuclear ERK signaling [113,133]. A recent study by Liu and colleagues elucidated a role for DUSP8 in cardiomyocyte growth, where overexpression resulted in eccentric remodeling and ventricular dilation due to myocyte lengthening and myocardial thinning. In DUSP8-null hearts subjected to pathological stress, ERK1/2 signaling was upregulated. Given that other MAPK modules were relatively unaffected, DUSP8 was deemed specific to the ERK compartment [134]. Other DUSPs have been identified to play less specific roles in the ERK and other MAPK modules. A recent study by Zhao and colleagues implicated DUSP26 in cardiac hypertrophy by modulating p38 and JNK signaling [135]. While ERK signaling was found unaffected, these data illustrate complex crosstalk between the MAPK modules and their regulatory components. Additional phosphatases play a role in regulating ERK signaling. Indeed, Src homology 2 domain containing tyrosine phosphatase 2 (SHP2) has also been identified as integral to physiological ERK signaling, where deletion elicits dilated cardiomyopathy [13]. Although future studies will be needed to further address phosphatase-specificity in ERK signaling, their role in regulating the cardiac landscape cannot be overstated. The ERK cascade, as discussed above, requires synergism between several components including scaffold proteins and phosphatases to transduce a membrane signal and elicit controlled cardiomyocyte hypertrophic changes (Figure 1).

### 4.4. ERK in Heart Failure Onset and Progression

The importance of ERK is clearly demonstrated by the consequences of aberrant ERK signaling for cardiac function in humans. The ERK1/2 module has been implicated in heart failure and other diseases linked to pathological hypertrophy. An autophosphorylation of ERK1 at Threonine 188 (T188) was discovered by Lorenz and colleagues to induce maladaptive hypertrophy in a pressure overload mouse model and in failing human hearts [87]. Another study found enhanced T188 phosphorylation in patients with rapid course aortic valve stenosis [136]. This modification was found to exert spatial regulation through nuclear localization of ERK to activate hypertrophic targets Elk1, MSK1, and c-Myc [136]. Interestingly, T188 does not influence kinase activity, which is facilitated through the phosphorylation of T183 and T185, but it solely impacts nuclear translocation. Thus, significant hypertrophy is only observed when both phosphorylations occur, as both nuclear and cytoplasmic events trigger cellular consequences [136]. Further, data suggest that the phosphorylation of T188 is requisite to nuclear translocation of ERK, therefore making the residue a promising site for therapeutic modulation. Indeed, T188A mutation does not impact physiological cardiac growth, suggesting that intervention would solely impact pathological hypertrophy [136]. It was recently shown that T188 autophosphorylation requires ERK–ERK interaction and that T188 is essential to nuclear ERK accumulation [137]. In line with the discussed findings, continuous pressure overload in vivo can induce a transition from adaptive to maladaptive hypertrophic remodeling, primarily by way of angiotensin II (Ang II) binding to GPCR AT1R. Ang II induces vasoconstriction, thus causing hypertension at high levels to cause cardiac aberration. High Ang II levels correlate to hypertrophic signaling and myocardial infraction [138,139]. Indeed, evidence suggests that angiotensin-converting enzyme (ACE) inhibitors slow heart failure progression and improve hemodynamics [140]. Though some molecular details remain unknown regarding the mechanism for ACE inhibition, the above studies illustrate an inextricable link between ERK signaling and hypertrophy.

Alteration in ERK activity has also been linked to hypertrophic cardiomyopathy (HCM). Ras mRNA expression has been shown to positively correlate to the severity of hypertrophy in HCM [141]. Furthermore, mutations in the ERK signaling cascade occur in congenital diseases called RASopathies. Among other phenotypes, these beget cardiac defects, where two specific disorders—Costello and LEOPARD syndromes—often involve HCM [142]. In mice expressing Noonan syndrome-associated mutated RAF-1 (L613V), eccentric hypertrophy and decompensation following TAC have been observed. Interestingly, treatment with MEK inhibitor Mirdametinib normalized these cardiac defects, suggesting a role for aberrant ERK signaling in HCM development [143]. Cardiomyocytes derived from LEOPARD patient induced pluripotent stem cells show higher sarcomeric organization and increased size, correlating to increased hypertrophy [144]. Specifically, germline mutations to B-Raf in Noonan, LEOPARD, and cardiofaciocutaneous syndromes yield atrial–septal defects and pulmonary stenosis [145]. B-Raf plays an integral role in cardiac homeostasis, as evidenced by the impairment of angiogenesis and increased myocyte apoptosis with B-Raf and MEK inhibition [146]. While molecular insights still require elucidation, it has been proposed that the administration of such inhibitors may elicit adverse cardiovascular outcomes by way of increased cAMP and PKA activity, thus upregulating K^+^ channel phosphorylation. As a result, myocardial repolarization is downregulated [147]. While such inhibitors have been indicated for the treatment of cancers, the data indicate a need to understand their cardiotoxicity and systemic impacts.

### 4.5. Cardiac Ramifications of Therapeutic ERK Inhibition

The development of drugs that selectively inhibit B-Raf for cancer therapy have been the subject of intense focus in recent years, with numerous clinical trials yielding several FDA-approved treatments. Certain cancers, such as melanoma, are associated with an activating mutation in B-Raf, thus upregulating downstream ERK signaling [148]. As discussed, the administration of such therapeutics can elicit adverse cardiovascular side effects. Further, several clinical trials implement combination therapy, whereby multiple drugs are dispensed to target various MAPK signaling components, increasing the potential for additional deleterious cardiac effects. This combinatorial approach is necessary because most patients receiving monotherapy treatment develop a resistance to the drug through compensatory upregulation of MEK, which can be relieved via MEK inhibitor treatment [149].

Three FDA-approved Raf inhibitors are dabrafenib, vemurafenib, and encorafenib, each of which has been noted to affect the heart [150,151,152]. Numerous clinical trials have tested the efficacy of these drugs, including COMBI-d and COMBI-v [150], coBRIM [151], and COLUMBUS [152]. The most common adverse event related to cardiovascular homeostasis noted in these studies is hypertension, with the incidence of all-grade hypertension reaching as high as 29% of all study participants in the COMBI trials [150], but decreased ejection fraction [150,151,152], QT prolongation [151,152], peripheral edema [150,151,152], and increased blood creatine kinase [151,152] have also been noted. Importantly, the experimental design of the coBRIM and COLUMBUS trials highlighted that the combination of Raf and MEK inhibitors frequently resulted in worsened cardiac outcomes than Raf inhibitor monotherapy alone. Using these two trials as an example in the case of hypertension, treatment with Raf inhibitor alone resulted in 8.1% [151] and 5.7% [152] of patients developing hypertension, respectively. When combining the same Raf inhibitor with an MEK inhibitor, these incidences jump to 15.8% [151] and 10.9% [152], highlighting an important drawback to combining these drugs. Taken together, this evidence makes it clear that the use of Raf and MEK inhibitors in the treatment of cancer must be carried out while paying careful attention to their effect on cardiac homeostasis.

## 5. Conclusions

Cardiac hypertrophy occurs in response to both physiological and pathological stimuli, where extracellular factor-mediated signal transduction results in cellular changes to cardiomyocytes. As an intracellular mediator of extracellular signals, ERK activation has been undoubtedly proven critical for the heart. Indeed, the above studies illustrate the involvement of ERK signaling in cardiomyocyte homeostasis and cardiac stress responses. Several studies have identified the assembly of macrocomplexes containing upstream ERK effectors as a means to modulate signal strength and duration [153]. Thus, greater understanding of protein–protein dynamics within the cardiac ERK module may expand our current understanding of ERK’s role in hypertrophic responses [154,155,156,157,158]. Further, profiling of the crosstalk between ERK, other MAPK signaling compartments, and additional cellular pathways are necessary to establishing potential therapeutic targets for cardiac pathologies. Nonetheless, ERK signaling is inseparable from cardiac responses and provides a springboard for exciting studies to be done.

## Figures and Tables

**Figure 1 biology-10-00346-f001:**
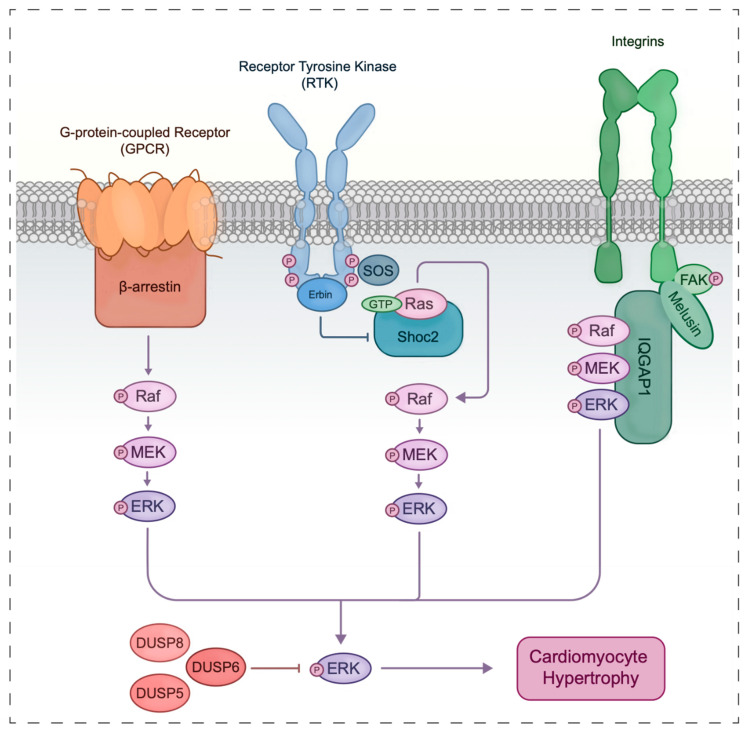
The cardiomyocyte ERK signaling event framework, where extracellular cues drive the activation of G-protein-coupled receptors (GPCRs), receptor tyrosine kinases (RTKs), and integrins to elicit ERK signaling and cardiomyocyte hypertrophy. For RTKs, the SOS exchange the SOS exchange factor activates Ras, scaffolded by Shoc2, to promote serial phosphorylation and downstream ERK activation. GPCR and integrin activation can occur under cardiomyocyte stress with respective scaffolding complexes (β-arrestin for GPCRs and the IQGAP1-Melusin-FAK supercomplex for integrins) functioning to temporally and spatially modulate the transduced signal. Active ERK functions to drive cardiomyocyte hypertrophy through the phosphorylation of cytoplasmic, cytoskeletal, and nuclear targets to modulate gene expression. Specific phosphatases such as DUSP5, 6 and 8 act to temporally limit ERK activation and favor signal termination.

**Table 1 biology-10-00346-t001:** Direct ERK interactors identified for both mouse and human. Targets were primarily obtained from a compendium of ERK targets [26].

Localization	Gene Name	Protein Name	Role/Function	References
Plasma Membrane	CAPN2	Calpain-2 catalytic subunit	Plays a role in epidermal growth factor (EGF)-mediated cell adhesion and motility.	[27,28,29]
	NOXA1	NADPH oxidase activator 1	Activates NADPH oxidase NOX1 to produce reactive oxygen species (ROS) that play a role in host defense, hormone biosynthesis, and sensing.	[29,30]
Cytoskeleton	AKAP12	A-kinase anchor protein 12	Anchors protein kinases A (PKA) and C (PKC) and plays a role in cell cycle regulation.	[29,31,32,33,34]
	AMPH	Amphiphysin	Plays a role in endoyctosis.	[29,35,36,37]
	MAP1B	Microtubule-associated protein 1B	Plays a role in neurite extension, where phosphorylation may induce microtubule association.	[29,32,33,34]
	TNKS1BP1	182 kDa tankyrase-1-binding protein	Functions in regulating the actin cytoskeleton.	[29,33,34,38]
Cytosol	DCP1A	mRNA-decapping enzyme 1A	Modulates gene expression through interaction with decapping enzyme Dcp2.	[29,33,39]
	RPS3	40S ribosomal protein S3	Functions as a member of the ribosomal small subunit; plays a role in DNA repair.	[29,32,33,34,40]
	RPS6KA1	Ribosomal protein S6 kinase alpha-1 (RSK1)	Acts downstream of ERK found in both the cytosol and nucleus.	[29,41]
	RPS6KA3	Ribosomal protein S6 kinase alpha-3 (RSK2)	Acts downstream of ERK found in both the cytosol and nucleus.	[29,41]
	RPS6KA2	Ribosomal protein S6 kinase alpha-2 (RSK3)	Acts downstream of ERK found in both the cytosol and nucleus.	[29,41]
	SPHK1	Sphingosine kinase 1	Activates proteins tied to proliferation and repression of apoptosis following activation through phosphorylation.	[29,42]
	STAT1	Signal transducer and activator of transcription 1-alpha/beta	Transduces interferon-, growth factor-, and cytokine-mediated signals.	[29,33,43,44]
	STIM1	Stromal interaction molecule 1	Upon intracellular Ca^2+^ depletion, translocates from the ER membrane to near the plasma membrane to activate store-operated calcium channels (SOCs).	[29,45]
	TP53	Cellular tumor antigen p53	Functions as a cell cycle regulator and tumor suppressor.	[29,46]
Nucleus	AHNAK	Neuroblast differentiation-associated protein AHNAK	May be required for neuronal cell differentiation; has been observed to disrupt PKC-PP2A complex to upregulate ERK signaling in NIH3T3 fibroblasts.	[29,32,34,47,48,49,50]
	ATF2	Cyclic AMP-dependent transcription factor ATF-2	Upon phosphorylation, upregulates genes tied to DNA damage response and cell growth.	[29,37,51]
	CEBPB	CCAAT/enhancer-binding protein beta	Phosphorylation of the CCAAT/enhancer-binding protein-b (C/EBPb) by ERK2 (not ERK1) enhances its interaction with SRF and its transactivation activity.	[29,52]
	ELK1	ETS domain-containing protein Elk-1	Primarily functions to transcribe c-Fos, which plays a role in cell proliferation and differentiation; phosphorylation enhances activity.	[29,37,53]
	EP300	Histone acetyltransferase p300	Regulates transcription via chromatin remodeling.	[29,54]
	ETS1	Protein C-ets-1	Regulates immune cell function.	[29,55]
	FAM195B	Mapk-regulated corepressor-interacting protein 1	Regulates co-repressor CtBP to mediate gene silencing during the epithelial-mesenchymal transition.	[29,34,56]
	FOXO3	Forkhead box protein O_3_	Regulates cellular processes, such as apoptosis and autophagy, where nuclear translocation occurs in response to stress.	[29,57]
	HNRNPH1	Heterogeneous nuclear ribonucleoprotein H	Mediates pre-mRNA splicing.	[58]
	JUN	Transcription factor AP-1 (c-Jun)	Plays an integral role in cell cycle progression.	[29,32,37]
	MCL1	Induced myeloid leukemia cell differentiation protein Mcl-1	Plays a role in regulating apoptosis.	[29,59]
	NDF1	Neurogenic differentiation factor 1	Regulates differentiation of neuronal and endocrine cells.	[29,37,60]
	NR5A1	Steroidogenic factor 1	Plays a role in the development of the primary steroidogenic tissues in both sexes.	[29,61]
	NUP153	Nuclear pore complex protein Nup153	Functions trafficking across the nuclear envelope.	[29,32,33,34,62,63]
	PRRC2A	Protein PRRC2A (BAT2)	Functions in oligodendrocyte specification; plays a role in pre-mRNA splicing.	[29,33,34,50,64]
	RPS6KA5	Ribosomal protein S6 kinase alpha-5 (MSK1)	Upon activations, functions to phosphorylate CREB1 and ATF1 in response to mitogenic or stress signaling.	[29,41]
	SMAD4	Mothers against decapentaplegic homolog 4	Balances atrophy and hypertrophy, where nuclear translocation occurs R-SMAD; Component of nuclear SMAD2/SMAD3-SMAD4 complex.	[29,37,65]
	TAL1	T-cell acute lymphocytic leukemia protein 1	Serves as a positive regulator of erythroid differentiation.	[29,37,66]
	TPR	Nucleoprotein TPR	Functions trafficking across the nuclear envelope; plays an integral role in cell division.	[29,32,34,50,67]
	XRN2	5’-3’ exoribonuclease 2	Functions in RNA degradation.	[29,32,34,68]

## Data Availability

This study did not involve Institutional Review Board protocols and did not generate new datasets.

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
