# Peer review of "ERK1/2: An Integrator of Signals That Alters Cardiac Homeostasis and Growth"

_biology, 2021, doi:10.3390/biology10040346_

Round 1
Reviewer 1 Report
This is a well-done, comprehensive review in the area of ERK signaling during cardiomyocyte function. I found the content to be very relevant (with historic as well as current citations) and interesting to those in the fields of cardiac function and kinase signaling.
Author Response
We thank this reviewer for their appreciation of our work.
Reviewer 2 Report
In this manuscript, Gilbert et al summarized comprehensively the recent proceedings of how ERK signaling regulates cardiac homeostasis and growth, which would have significant impact on future studies in this field. Overall, it is an excellent review manuscript, but still can be further improved before publication. Here, I have three points below to help authors to improve their manuscript:
- Germline mutations in RAF family kinases cause cardiac developmental disorders. This is tightly related to the theme of this manuscript. Authors should discuss all related discoveries and their underlying mechanism in manuscript.
- One of important proceedings in the field of ERK signaling is the understanding of activation mechanism of RAF/MEK/ERK kinase cascade (Santos and Crespo, Sci Signal., 2018). A discussion within context of cardiac homeostasis and growth will significantly elevate the impact of this manuscript.
- Development of RAF inhibitors for cancer therapy is a very hot topic in current biomedical research. Do these inhibitors approved for clinic usage have any effects on cardiac homeostasis and growth ? if do, authors may briefly discuss in their manuscript.
Author Response
We thank this reviewer for their work in helping us improve this manuscript. As suggested, we now discuss the importance of RAF mutations as well as the effects of RAF inhibitors on the heart. We also now discuss the importance of understanding how protein-protein interactions upstream of ERK influence this signaling pathway. All the revisions made are tracked in the manuscript (pages 18-20). We appreciate this reviewer's advises.